# Cycle-Informed Triaxial Sensor for Smart and Sustainable Manufacturing

**DOI:** 10.3390/s25144431

**Published:** 2025-07-16

**Authors:** Parisa Esmaili, Luca Martiri, Parvaneh Esmaili, Loredana Cristaldi

**Affiliations:** 1Dipartimento di Elettronica, Informazione e Bioingegneria, Politecnico di Milano, I-20133 Milan, Italy; luca.martiri@polimi.it (L.M.); loredana.cristaldi@polimi.it (L.C.); 2Department of Computer Engineering, Cyprus International University, Northern Cyprus, via Mersin 10, 99258 Nicosia, Türkiye; pesmaili@ciu.edu.tr

**Keywords:** predictive maintenance, empirical mode decomposition, triaxial accelerometer, vibration analysis, CNC machining, condition monitoring, Industry 5.0, smart manufacturing, embedded systems

## Abstract

Advances in Industry 4.0 and the emergence of Industry 5.0 are driving the development of intelligent, sustainable manufacturing systems, where embedded sensing and real-time health diagnostics play a critical role. However, implementing robust predictive maintenance in production environments remains challenging due to the variability in machine operations and the lack of access to internal control data. This paper introduces a lightweight, embedded-compatible framework for health status signature extraction based on empirical mode decomposition (EMD), leveraging only data from a single triaxial accelerometer. The core of the proposed method is a cycle-synchronized segmentation strategy that uses accelerometer-derived velocity profiles and cross-correlation to align signals with machining cycles, eliminating the need for controller or encoder access. This ensures process-aware decomposition that preserves the operational context across diverse and dynamic machining conditions to address the inadequate segmentation of unstable process data that often fails to capture the full scope of the process, resulting in misinterpretation. The performance is evaluated on a challenging real-world manufacturing benchmark where the extracted intrinsic mode functions (IMFs) are analyzed in the frequency domain, including quantitative evaluation. As results show, the proposed method shows its effectiveness in detecting subtle degradations, following a low computational footprint, and its suitability for deployment in embedded predictive maintenance systems on brownfield or controller-limited machinery.

## 1. Introduction

The ability to diagnose faults, optimize maintenance strategies, and extend equipment lifespan is becoming increasingly vital as industries move towards more sustainable manufacturing practices. Robust fault diagnosis and status signature analysis (SSA) techniques [1,2] play a key role in ensuring the reliability and efficiency of complex systems. In the fields of additive and subtractive manufacturing, the condition monitoring of computer numerical control (CNC) machining centers is a cornerstone of predictive maintenance strategies in smart manufacturing. As manufacturers increasingly demand high precision and waste-less production linked directly to optimizing machining environments [3], various SSA methods have been proposed in the literature [4,5]. In thermal monitoring, we can detect excessive heat due to friction, misalignment, or bearing failure, which can be detected using infrared thermography or contact temperature sensors providing warning signs, especially in rotating or high-load machinery [6]. Analyzing the changes in stator current harmonics such as by using motor current signature analysis (MCSA) is the developed approach in electrical-signal-based techniques, which are widely used to diagnose faults in induction motors by [7,8]. Alternative non-contact methods such as acoustic emission (AE) sensors [9] capture high-frequency transient waves associated with crack formation, tool wear, or delamination in composites, offering a non-intrusive and sensitive means of monitoring localized structural changes. Such approaches are commonly used alongside other diagnostic techniques like thermal analysis and electrical signal analysis. Vibration analysis is a fundamental technique in fault diagnosis, especially for rotating machinery. During machining, vibrations can result from insufficient rigidity in the machine, errors in the workpiece–tool connection, or changes in cutting conditions throughout the process [10].

In terms of non-destructive vibration measurements and an alternative to the traditional complex and costly interferometers, self-mixing interferometry (SMI) offers compact, economical, and highly sensitive solutions, enabling a broad range of applications [11,12,13,14]. Despite its compact design, they often require dedicated hardware. Accelerometers are favored for their high bandwidth, sensitivity, and ease of integration into automated monitoring systems in several applications [1]. Micro-electromechanical system (MEMS)-based vibration sensors, which are small, low-power, and cost-effective, are increasingly integrated into Industrial Internet of Things (IIoT) frameworks to enable smart maintenance strategies [15,16]. A low-cost triaxial capacitive MEMS accelerometer, combined with a Hall-effect sensor, has been proposed in [17] for measuring vibration and leakage flux in induction motors. A cost-effective triaxial accelerometer is developed in [18], utilizing three semicircular hetero-core optical fibers for a precise triaxial acceleration measurement. This design induces optical loss through mechanical deformation while maintaining a low transverse sensitivity. Considering industrial use-cases and their current transition from Industry 4.0 to Industry 5.0 represents a significant shift in how industrial systems are designed, maintained, and optimized.

While Industry 4.0 focused on automation, cyber-physical systems, and data-driven decision-making [19], Industry 5.0 emphasizes the reintegration of human intelligence into AI-driven processes, fostering a collaborative relationship between humans and machines. Such a collaboration helps us to correctly interpret the system’s response in order to identify and address biases in the model, enhancing performance. Recently, Artificial Intelligence (AI) and Machine Learning (ML) have become vital tools for enhancing malfunction signature analysis [20]. For example, advanced deep learning techniques like the RoughLSTM-based approach for detecting vibration anomalies [21] show how effectively neural networks can model temporal patterns in CNC signals. In [22], a large-scale AI benchmarking study evaluated 36 machine learning and deep learning models. These studies include convolutional networks like ResNet and InceptionTime, as well as recurrent networks like LSTM and BiLSTM. The findings demonstrate that deep convolutional models consistently deliver a high classification accuracy, underscoring the expanding role of data-driven AI in smart manufacturing.

In addition, hybrid methods combining multiple AI and Machine Learning models have gained popularity for fault diagnosis in sensor networks, as they leverage the strengths of different models to enhance the overall fault detection performance [23,24,25,26]. Although these methods show reliable results, they may face challenges in real-world production environments over extended periods, considering factors like feature drifts and varying tool operations. In addition, industrial constraints frequently limit the installation of multiple or high-end sensors, due to both economic considerations and physical space limitations within the machine tool structure.

Maintenance accessibility and system retrofitting all introduce practical barriers to widespread sensor deployment. This underscores the importance of developing methodologies that can extract maximum diagnostic value from minimal sensing input, where even a single vibration signal can be leveraged to reveal detailed process dynamics, potential anomalies, and early indicators of failure. To address these gaps, this paper introduces a lightweight, embedded-compatible vibration analysis framework centered around a single triaxial MEMS accelerometer. In this paper, our focus is to tackle a more fundamental and often overlooked challenge in real-world and dynamic predictive maintenance schemes. Traditional fixed-length segmentation approaches do not account for actual machine dynamics and often result in misaligned or cycle-incomplete segments, diluting the diagnostic relevance and potentially mixing signals from different operational states. In this paper, the core of the contribution lies in a cycle-synchronized segmentation method, which leverages velocity profiles extracted directly from the accelerometer signal, consequently, avoiding the misinterpretation of dynamic vibration data and enabling a causally coherent view of each machining cycle. This addresses a critical research gap in the literature, where most vibration-based monitoring frameworks assume access to controller signals or static segmentation windows. Unlike existing methods, the proposed approach is designed to operate in controller-limited environments and enables the extraction of interpretable, physically grounded health signatures that can serve as reliable input for downstream classification or prognosis. This novelty is particularly relevant for embedded monitoring applications where cycle awareness, a low computational overhead, and explainability are key requirements. By applying low-pass filtering and cross-correlation, the method identifies operational cycles without requiring any control signal or encoder feedback. This preserves physical causality and operational context—something often lost in fixed-length windowing approaches. Following segmentation, each cycle is subjected to an individual application of Empirical Mode Decomposition (EMD). In contrast to fixed-basis transformations like Wavelet Transform or Short-Time Fourier transforms (STFTs), EMD adaptively breaks down signals into Intrinsic Mode Functions (IMFs) according to local signal properties. This works especially well for detecting nonstationary, minor events like early tool wear or misalignment, which conventional transforms or envelope detection methods would miss. The following are this study’s primary contributions:It introduces a dynamic cycle-informed segmentation technique eliminating the need for controller access and utilizing only accelerometer-derived information to match the vibration data with actual machining cycles.It develops an EMD-based decomposition pipeline that captures interpretable and cycle-resolved health indicators tailored for CNC environments.In line with Industry 5.0 principles, it presents a hardware-efficient design paradigm that supports deployment in embedded systems, thus enabling scalable predictive maintenance across legacy and controller-limited machines.

Using an industrial benchmark acquired from real-world CNC operations involving different tool operations with labelled healthy and faulty processes, this paper primarily presents the validation of the proposed method on long-term data [27]. The results show the proposed approach’s ability to differentiate not only between OK and NOK operations but also to detect early degradations and subtle shifts in vibration signatures indicative of emerging faults. Although the proposed method is evaluated solely on CNC-milling processes with a relatively low sampling rate of 2 kHz, it is broadly applicable to other industrial systems exhibiting periodic behavior. This comprises automated assembly lines, injection molding systems, and robotic arms used in pick-and-place tasks. In such cases, even without access to controller data, these systems generate repeatable mechanical cycles that produce characteristic vibration patterns. The proposed approach is well-suited to monitoring these processes, particularly in brownfield or controller-limited environments where internal machine signals are unavailable, but vibration data can still be captured externally. The rest of the paper is organized as follows: Section 2 details the workflow of the proposed approach, Section 3 discusses the results obtained from a real-world manufacturing benchmark for different status signatures representing silent anomalies, and Section 4 presents the conclusion and future steps.

## 2. Materials and Methods

CNC-milling machines, essential in industrial manufacturing, frequently work in demanding environments, introducing significant variability in mechanical loads, cutting forces, and vibration patterns. Consequently, traditional fixed-window segmentation methods used in vibration-based fault diagnostics often fail to align with actual machine operations, causing ambiguous signal interpretations and prediction in product quality. In contrast, the framework proposed in this paper focuses on cycle-synchronized signal segmentation and interpretable status signature extraction—designed to complement future machine learning models with physically grounded, process-aware inputs. The aim of this paper is to propose a dynamic, low-cost and embedded solution to integrate the dynamic process without additional hardware minimizing sensing input while extracting maximum diagnostic parameters. As illustrated in Figure 1, the system is based on a small, triaxial accelerometer that is integrated into the spindle housing of a CNC-milling machining center utilizing a smart sensor framework. The sensor selection is driven by a balance between bandwidth, resolution, and power consumption, aiming to enable continuous monitoring of structural and tool-related dynamics with minimal system intrusion.

With a bandwidth of more than 1 kHz, the sensor can detect a wide range of vibrations caused by machine-level disruptions as well as tool–workpiece interactions. It is sensitive to both stationary and transient problems, such as chatter, misalignment, and im-balance, as a result of its low noise floor and dynamic range. From a system-level perspective, the sensor does not function as a passive transducer but is conceptualized as a smart sensing unit capable of autonomous signal processing. The whole workflow in the current implementation is designed with embedded deployment in mind, even though signal processing is carried out offline. The proposed approach combines the sensor with a microcontroller that can extract features at the edge, segment cycles, and filter in real time. Without requiring interfaces with machine control logic or extra instrumentation, this design paradigm makes it easier to develop a modular, inexpensive sensor node that can be used in controller-limited or legacy systems. The framework performs better than fixed-window and frequency-limited techniques in accurately extracting failure-related patterns by matching derived segments with real operational cycles. It also integrates seamlessly with learning architectures for online classification and prognosis. Figure 2 shows the proposed framework for health signature extraction, which is structured in two stages: adaptive cycle segmentation and intrinsic mode decomposition.

The first stage ensures that only complete and contextually relevant machining cycles are analyzed. This is achieved by filtering and aligning triaxial accelerometer signals based on periodic motion patterns identified through cross-correlation. By doing so, the system discards transient data unrelated to the actual operational cycle, mitigating the impact of irregularities like idle time, clamping, or tool changes. Therefore, instead of introducing another classifier, in this paper, we concentrate on developing interpretable signal decomposition aligned with real process cycles. It lays the groundwork for creating more comprehensive, multi-class annotated datasets. Algorithm 1 presents the pseudo-code of the proposed cycle-informed segmentation-based status signature.

**Algorithm 1:** Cycle-Informed Segmentation and EMDFunction decompose($a_{1}$(*t*));
**Input:** Triaxial acceleration signal *a*(*t*)
**Output:** Intrinsic Mode Functions (IMFs) per segmented cycle
1. Filter *a*(*t*) to obtain $a_{1}$(*t*)
2.    Integrate $a_{1}$(*t*) to compute velocity profile $S_{1}$(*t*)
3.    Select a reference cycle segment $S_{2}$(*t*)
4.    Compute cross-correlation between to detect cycle boundaries
5.    Extract acceleration signal segments $C_{1},C_{2},\ldots ,C_{3}$ based on detected cycle intervals
6.    for each cycle segment $C_{i}$ **do**
        a.  Initialize residual: *r_0_ ←* $C_{i}$ *ᵢ*
        b.  **while** $r_{i}$ is not monotonic **do**
              i.  Extract upper envelope $e_{max}(t)$ and lower envelope $e_{min}(t)$

              ii.  Compute mean envelope: m(*t*) *=* ($e_{max}(t)$ *+*
$e_{min}(t)$)/2
              iii.  Subtract mean: *h*(*t*) *←* $r_{i}$ *− m*(*t*)
              iv.  **if** *h*(*t*) satisfies IMF criteria **then**
                  – Store IMF: ${IMF}_{i}$ ← *h*(*t*)
                  – Update residual: *r_i_*_+1_ ← *rᵢ* − *IMFᵢ*
              **else**
                  – Set *rᵢ* ← *h*(*t*) (continue sifting)
              **end**
        **end**
7**.    end**

### 2.1. Cycle-Informed Segmentaion

Due to the low-frequency nature of the axes’ motion, a lowpass filter can be applied to the acceleration signal to extract effective information regarding actual movement even without access to the controller. The extraction of a single machining cycle begins with preprocessing the triaxial accelerometer data using a low-pass Butterworth filter. This step isolates the motion-induced vibration signature from background noise. The filtered signals are then cross-correlated with a reference cycle template to identify peak alignments that signify cycle boundaries. The indices corresponding to high similarity scores are selected to segment the input signal. To validate the extracted cycle, velocity profiles along X, Y, and Z axes are normalized and visually inspected. This alignment step is crucial in maintaining phase coherency across samples. The segmented cycle is later used as the reference window for status signature extraction, ensuring repeatability across all other samples.

This approach eliminates ambiguities arising from arbitrary or fixed windowing approaches. The transfer function of the Butterworth infinite impulse response filter in the frequency domain is represented by (1), where $w_{c}$ is the cutoff angular frequency and n is the order which is determined by the steepness parameter. Here, to attenuate high frequencies more aggressively, steepness = 0.95, and, following (2), the filter order is 4.(1)$$H\left(s\right)=\frac{1}{1+{\left(\frac{s}{w_{c}}\right)}^{n}},$$(2)$$n=\left[-\frac{\log(\;1-0.95)}{log2}\;\right]\;≅4,$$

The cut-off frequency used in the low-pass Butterworth filter was 50 Hz, which was selected to preserve process-level motion dynamics. Figure 3 shows typical time signals acquired by a triaxial sensor mounted on CNC machine while operating end-mills [27] in addition to the filtered signal.

Since the filtered signal is associated with the axes’ motion, the speed profile can be extracted by integrating the filtered acceleration signals over time to obtain the velocity in the x, y, and z measuring axes as shown in Figure 4.

Although a small offset in acceleration will affect the velocity, the single machining cycle can still be retrieved easily. As shown in this figure, combining the repeated movement observed in speed profile of y and z axes is enough to extract the reference signal using x axis. Following the workflow represented in Figure 2, once the preprocessing step for both the input and the reference signals is performed, the next step is to evaluate cross-correlation using (3) and identify the corresponding index for signal segmentation:(3)$$R_{xx\;}=\sum_{l=-N}^{N}\sum_{n=0}^{N-1}s_{1}\left(n\right)\times s_{2}(n-l),$$
where $s_{1}$ represents the input signal and $s_{2}$ is the reference signal associated with OP10. The normalized amplitude of the result of cross-correlation ($R_{xx\;}$), with peak identification shown in Figure 5. As expected, the algorithm automatically discards the first and last part of the time signal labeled as “identified indices” since they are irrelevant to the operation that the machine executes. The verified indices, instead, are those associated with the selected operation and correspond to the ideal time of the single operating cycle. The result of the segmented time signal associated with the single operation cycle is demonstrated in Figure 6. Once the coherency between the acquired signals and the operating cycle is guaranteed, $A_{swc(t)}$ are the input of the second stage to extract IMFs representing oscillatory mode.

### 2.2. Stage 2: IMF-Based Status Signature

Status signatures refer to unique data patterns that characterize the operational state of a system. Traditional signal processing techniques such as Short-Time Fourier Transform (STFT), Wavelet Transform (WT), and envelope detection have been widely employed in the condition monitoring of rotating machinery and CNC systems. STFT provides a time–frequency representation by segmenting the signal into short-time windows; however, its fixed window size imposes a trade-off between time and frequency resolution, making it less effective for analyzing signals with rapidly changing spectral content.

Wavelet Transform offers multi-resolution analysis, allowing better localization of transient features in signals. Nevertheless, its performance heavily depends on the choice of the mother wavelet, which may not optimally match all signal characteristics. Envelope detection is a simple technique effective for identifying amplitude modulation caused by bearing faults. However, it lacks frequency localization and is sensitive to noise, limiting its diagnostic capabilities in complex industrial environments. In contrast, Empirical Mode Decomposition (EMD) [28] is a fully data-driven and adaptive method that decomposes nonstationary and nonlinear signals into Intrinsic Mode Functions (IMFs) without the need for predefined basis functions. This adaptability makes EMD particularly well-suited for extracting process-specific status signatures from complex vibration signals in industrial CNC environments, where variability in cutting dynamics and machine loading leads to nonstationary behavior. By preserving both temporal and spectral characteristics of each mode, EMD enhances the interpretability and diagnostic value of the extracted features. Different versions of EMD such as Ensemble Empirical Mode Decomposition (EEMD) [29] have been developed to reduce mode mixing by introducing white noise to the original signal and performing multiple EMD decompositions. The resulting IMFs are then averaged across these ensembles. An improvement upon EEMD, Complete Ensemble Empirical Mode Decomposition with Adaptive Noise (CEEMDAN) [30], adds adaptive noise at each decomposition stage, leading to improved reconstruction and less residual noise.

Despite these advances, many industrial studies still rely on artificial fixed-length segmentation [31,32,33,34], which can obscure meaningful health indicators during cycle transitions. This work advances the state of the art by introducing cycle-aware EMD for vibration data segmentation considering edge-computing potentiality. As presented in the proposed algorithm in Figure 2, once the signal is segmented, Empirical Mode Decomposition (EMD) is applied to each working cycle individually. EMD is chosen for its ability to handle nonlinear and nonstationary data without requiring predefined basis functions. Through an iterative sifting process, each segment is decomposed into a finite set of Intrinsic Mode Functions (IMFs) and a residual trend. Lower-order IMFs capture high-frequency machine–tool interactions, while higher-order IMFs reflect global vibrations caused by load variations or system imbalances.

Unlike fixed-size window methods that segment data arbitrarily, our approach extracts vibration profiles tied to actual cycle boundaries, preserving causality. The amplitude spectra of selected IMFs are analyzed to detect shifts in frequency content, harmonic imbalance, and peak intensities that correlate with process degradation. This process-aware decomposition enhances sensitivity to early faults and enables cycle-level anomaly tracking. Using the extracted effective profile of the operation and measuring the correlation, the acquired signal is divided into sub-signals ($A_{swc(t)})\;$to ensure the extracted data corresponds to relevant single working cycles. This establishes a reliable monitoring system where the features are extracted or evaluated always on the same monitoring window defined by the actual working cycle while discarding irrelevant data. The second block breaks down each $A_{swc(t)}$ into Intrinsic Mode Functions (IMFs), which help to analyze nonstationary or complex signal behaviors. The core principle of EMD lies in decomposing a complex signal into a set of Intrinsic Mode Functions (IMFs),${\;c}_{i}\left(t\right),\;$and a residual trend ($r_{n})\;$as shown in (4).(4)$$x\left(t\right)=\sum_{i}^{n}c_{i}\left(t\right)+r_{n}\left(t\right),$$

The sifting process is a key operation in EMD that isolates each Intrinsic Mode Function (IMF) from the original signal in an iterative manner. It works by removing low-frequency trends and refining high-frequency oscillations, ensuring that each IMF represents a simple oscillatory mode. An IMF must meet two criteria: As the first criteria, the number of extrema and zero crossings differ by at most one. Secondly, the mean envelope m(t) is sufficiently close to zero. The first IMF captures the highest frequency content. The second IMF also contains high-frequency components but is broader in range compared to IMF 1. Meanwhile, the third and fourth IMFs primarily capture lower-frequency content, mostly below 100 Hz. Figure 7 shows the spectrum of each extracted IMF, which is used in status analysis in this paper and can be combined with features from the time domain.

As shown in this figure, low-order IMFs carry high-frequency spectral components, whereas the higher order presents lower-frequency ones. The detected peak at 250 Hz in IMF1 and 2 corresponds to the rotation speed of 1500 RPM.

## 3. Results: Real-World Manufacturing Benchmark

CNC-milling machines, essential in industrial manufacturing, frequently work in demanding environments and are prone to issues such as tool breakage and poor clamping. Due to the tough working conditions and varying loads, the advanced algorithms developed for these machines are often restricted in practical use, as they are generally trained on data gathered in controlled lab environments over short time spans. To address this limitation, [27] introduces a benchmark dataset compiled over two years from real-world CNC machine operations. The proposed methodology offers a unique area for addressing real-world machining process research. Through an investigation on the publicly available industrial benchmark dataset [27], the performance of the methodology is validated on a practical basis. The key reason for selecting such a benchmark lies in its long-term coverage and realistic variability features. In addition, binary labeling is provided based on the inspection of an expert, which provides us with a rare ability to evaluate the segmentation and diagnostic algorithms under limitations. Finally, while the majority of available studies are simulation-based or laboratory-condition studies, this benchmark reflects real operational values such as operational noise, tool wear, and cycle diversity. The experimental setup includes a triaxial accelerometer aligned with the linear motion axis of the machine and mounted on the rear end of the spindle housing of the horizontal four-axis CNC-milling machine as shown in Figure 8.

The data is sampled at a frequency of 2 kHz, with no additional information available from the controller. The dataset consists of 15 distinct tool operations, labeled OP01–OP15, representing a sequence of various tasks with specific characteristics, including different speeds, feed rates, durations, and operation types such as end mills, straight, and drilling. In high-speed machining operations, tools must be regularly mounted and removed from the spindle chuck, which can occasionally lead to operational failures. These failures are typically due to factors like tool misalignment, chip accumulation, improper clamping, or tool breakage. To maintain a high product quality, an expert on the shop floor performs post-batch inspections at a gauging station, evaluating the finished workpieces and classifying the process health as either OK (healthy) or NOK (faulty).

### 3.1. IMFs and Cycle-Informed Signature Analysis: OK vs. NOK Processes

Following EMD, the first four IMFs of each cycle are subjected to a spectral analysis. IMF1 and IMF2 typically represent high-frequency dynamics, while IMF3 and IMF4 capture lower-frequency oscillations linked to the machine state. Figure 9 illustrates the time-domain signals captured across the three axes for the annotated NOK process and while extracting single working cycles derived through the outlined framework as shown in Figure 2.

As shown in Figure 8, ${\;A}_{swc(t)}$ consists of only two complete cycles. This limitation arises due to the constraints in signal acquisition, where data recording commenced midway through the first operational cycle and concluded before the completion of the final cycle. Consequently, only two full cycles could be effectively reconstructed from the available data. The comparison of the spectral characteristics of the extracted IMFs for the NOK and OK-identified processes is shown in Figure 10. In this figure, the test classified as a NOK process is compared with a healthy process. The four subplots correspond to different IMFs, representing progressively lower-frequency components. We can underline distinct peaks around 250 Hz corresponding to the rotational speed but which are more pronounced in the NOK process, suggesting an anomaly or fault-induced excitation. The third and fourth IMFs primarily exhibit lower-frequency content, with the NOK process displaying a broader spectral distribution compared to the OK process. While a time-domain analysis in examining two OK and NOK processes allows for a straightforward differentiation between them, a deeper examination of the spectral distribution of IMFs offers additional valuable insights, such as localized frequency shifts, harmonic excitations, or resonant behaviors associated with faults that may not be immediately apparent in the time domain. However, the aim of this paper is not only to extract the status signature of the distinguishable faulty and healthy processes, but also to address challenges like the one associated with fast position changes that can cause discrepancies within the process.

Acknowledging the difficulty of obtaining sufficient labeled data in this industrial context, a two-stage learning method was proposed in [31], utilizing a prototypical few-shot learning approach. As the results show, it reduces the reliance on large, labeled datasets and achieves an F1-score of 90.3%. In the first stage, an autoencoder is used to learn features from unlabeled data, and, in the second stage, these features are fine-tuned using a small amount of labeled data through a prototypical network. In [32], various Machine Learning (ML) models and Wavelet Packet Transform (WPT)-based feature extraction techniques are introduced to detect anomalies. This involves evaluating 50 mother wavelets, two decomposition levels, and five ML models (Random Forest, Support Vector Machine, Multilayer Perceptron, Convolutional Neural Network, and LightGBM).

The classification accuracy was reported as an F1-score of 95%, while emphasizing the influence of mother wavelet selection on classification performance. Despite the impressive classification results, both studies mainly concentrate on binary classification (OK vs. NOK machining processes). While binary classification can generate maintenance alerts, failure diagnosis is necessary for taking the appropriate actions. This leads to one of the primary challenges in this industrial benchmark, as highlighted in [27], regarding segmenting time series using fixed-length windows (4096 points).

### 3.2. Improving Segmentation Fidelity

As noted earlier, most studies using this dataset train various machine learning models to distinguish between OK and NOK cycles. Although useful, these approaches typically depend on fixed-length windows or require exact manual segmentation. But these conditions are rarely practical in real-world production environments. Our approach overcomes this limitation by providing an embedded-friendly solution that automatically aligns signals with actual machine cycles. The proposed method tries to use only accelerometer data, without needing control logic or encoder inputs. This physically grounded, repeatable segmentation improves the accuracy and clarity of the extracted status signatures, creating a reliable foundation for future classification and machine learning applications. In operations, especially with fast position changes, OK and NOK classes may become nearly indistinguishable in mid-cycle regions using fixed windows. To demonstrate the benefit of cycle-informed segmentation, we compared the extracted signals with those obtained using 4096-point fixed windows. Figure 11 shows the time-domain acceleration signal including the filtered signal while the machine is operating OP08, with a rotation speed of 15,000 RPM, feed rate of 50 mm/s, and duration of about 37 s for both the OK and NOK processes. Considering the time-domain signals, the difference in the executing program is notable following the filtered signal where a small choice of window size from the middle of operations exhibits identical characteristics in both the OK and NOK classes. The resulting status signature of both tests is represented in Figure 12.

As shown in this figure, the proposed method shows distinguishable status signatures without any overlapping. NOK processes exhibit higher amplitudes in the first and second harmonics, more pronounced second harmonics to compare with OK process, and more erratic IMF 3 and IMF 4 behavior. Latent degradation and anomalies are also revealed using a harmonic ratio analysis across the IMF spectra. For example, an increase in the ratio of the second to first harmonic in IMF1 suggests the emergence of nonlinear dynamic effects.

### 3.3. IMFs and Status Signature Analysis for Early Detection

Ensuring quality control of the produced workpieces at a gauging station on the shop floor during production is a highly challenging task. In certain processes, accurate annotations may be lacking due to the manual effort required for gauging, or the quality may appear acceptable while a hidden pattern is in developing stage. In the following section, we assess the effectiveness of the proposed approach in addressing some of these hidden scenarios.

#### 3.3.1. Potential Failure

As the tool experiences wear, energy levels in the lower-frequency bands increase due to the growing instability of the cutting process. This leads to a higher amplitude of low-frequency components, which can be effectively captured in the frequency spectrum through higher IMFs. To assess the effectiveness of the proposed method, we analyze its ability to detect such developing anomalies. Figure 13 shows an example of such developing anomalies by providing not only alerts but also more details on the nature of such anomalies. The combination of IMF1 (sharp peaks) and IMF3 (energy spread) provides the best discriminative power for early fault detection. The OK–Potential Failure case clearly deviates from OK in both IMF2 and IMF3, suggesting this method is effective for catching silent degradation. The repeated 250 Hz peak in NOK and PF further validates the methodology’s sensitivity to real operational faults. It should be noted that the detected potential for a failure process was annotated as an OK process.

#### 3.3.2. Potential Malfunctioning

Harmonics are integer multiples of the fundamental frequency, and the ratio between the first harmonic (fundamental frequency) and the second harmonic, the harmonic ratio, offers valuable insights into a machine’s vibration characteristics and overall health. Figure 14 represents a comparison between the resulting status signatures for two OK processes where one carries valuable information on tool malfunctioning. In the OK-labeled process, as observed in IMF1 (depicted with a thinner line width), the first harmonic at 250 Hz is distinctly visible with a sharp peak, while the second harmonic at 500 Hz is also present but appears less prominent than the first. As a result, the first harmonic remains dominant over the second, characteristic of a healthy machine where vibrations primarily occur at the fundamental frequency, lowering the harmonic index. However, in the potentially malfunctioning process, the second harmonic becomes more pronounced compared to the OK process, and the increasing ratio between the first and second harmonics further indicates the presence of nonlinearities in the system integrity. Notably, the dominance of the second harmonic is evident in the NOK-classified process, as illustrated for OP08 in sub-Section 3.2.

## 4. Discussion

In this paper, we introduced a novel cycle-informed segmentation and IMF-based decomposition framework that enables meaningful and interpretable health status signatures from vibration data even in the absence of controller feedback, a condition commonly encountered in brownfield or controller-limited industrial systems. Many existing works of literature apply segmentation without ensuring the alignment to true machining cycles. This leads to misaligned or incomplete segments related to machine cycles resulting in a non-realistic diagnostic value and mixed signals from different operating states. Compared to fixed-window methods, our approach consistently extracts cycle-resolved segments that better align with physical process boundaries, thereby preserving causal relationships and improving the reliability of extracted features. To effectively address such gaps, we focused on data from real-world scenarios where it reflects real process variability, operational dynamics, presence of noise, binary labeling on final quality, and a lack of controller data and a lack of historical data, characteristics commonly found in brownfield or controller-limited industrial systems. This aligns well with our paper’s goal: to propose an interpretable, embedded-compatible method that works even under such constraints. In Table 1, we summarized the three most challenging case studies, while evaluating the key observations and effectiveness of the proposed algorithm. As shown in this table, the proposed method demonstrated a high sensitivity to evolving tool–workpiece interaction, with the ability to detect latent anomalies before they become critical. Furthermore, the framework remains lightweight and well-suited for embedded deployment. It requires no model retraining or parameter tuning and relies only on cycle-adapted signal filtering, integration, and cross-correlation, making it compatible with resource-constrained platforms. This design supports controller-independent, scalable predictive maintenance systems and aligns with the emerging goals of Industry 5.0.

## 5. Conclusions

The reliability of any classification or prognosis system depends heavily on the quality and contextual alignment of the input features, especially in a dynamic environment. In this paper, an adaptive approach is proposed for extracting status signatures from real-world machining data. By enhancing the accuracy of status signature extraction and eliminating issues inherent in fixed-window segmentation, this method promises to significantly improve the effectiveness of real-time fault detection systems in industrial environments. By combining low-cost vibration sensing with cycle-synchronized decomposition, the proposed system can operate without direct access to the machine controller, making it ideal for brownfield retrofitting and controller-limited systems. In this paper, the proposed algorithm is tested on a real-world industrial benchmark including workpiece-quality-based annotated healthy and faulty processes. The resulting intrinsic mode functions as represented in this paper exhibit distinct and cycle-specific harmonic patterns that enable a clear binary classification and the detection of developing failure, highlighting the method’s capacity to handle early-stage degradation while providing insightful information for predictive maintenance.

The future FPGA/MCU integration and scalability to multi-axis or multi-sensor configurations are made possible by this co-design between the triaxial sensor placement and the segmentation method, which guarantees a minimum dependency on external data while lowering the computational load. Although the present study focuses on software-based validation using triaxial vibration data acquired at 2 kHz, each algorithmic block—cycle-synchronized segmentation, empirical mode decomposition (EMD), and spectral feature extraction—was benchmarked with the computational cost in mind. The segmentation algorithm relies only on low-pass filtering and cross-correlation, both of which are efficient and can be executed on mid-range microcontrollers. In addition, the proposed workflow avoids memory-intensive transforms and fixed-length buffers, making it compatible with memory-constrained systems. In terms of robustness, the use of low-pass filtering and the velocity-domain correlation makes the approach resilient to high-frequency noise and minor transient disturbances. Moreover, the method is tolerant to a reasonable variation in the sensor-mounting orientation, as long as the primary motion axes are captured.

While the proposed method demonstrates a strong potential for interpretable signal segmentation and fault signature extraction in industrial machining, certain limitations outline opportunities for future enhancement. The framework assumes a degree of cycle periodicity, which aligns well with many industrial operations but may require adaptation for processes with a highly irregular timing or non-repetitive tool paths. The current sampling rate (2 kHz) is sufficient for process-level dynamics but may not capture micro-scale or high-frequency faults such as localized early-stage bearing defects. Such further improvements can lead to a smart and sustainable digital twin enabler where these cycle-resolved features are integrated as a foundation for future machine learning models where fault tolerance and early failure detection are crucial.

## Figures and Tables

**Figure 1 sensors-25-04431-f001:**
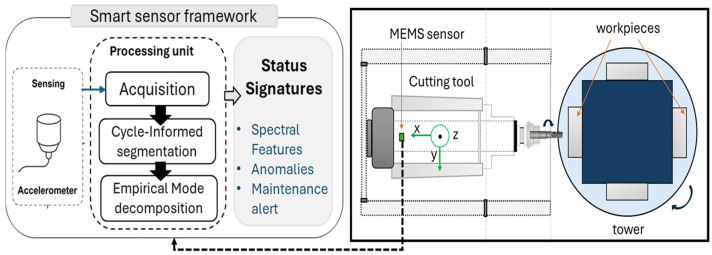
Schematic sketch of the proposed smart sensor framework.

**Figure 2 sensors-25-04431-f002:**
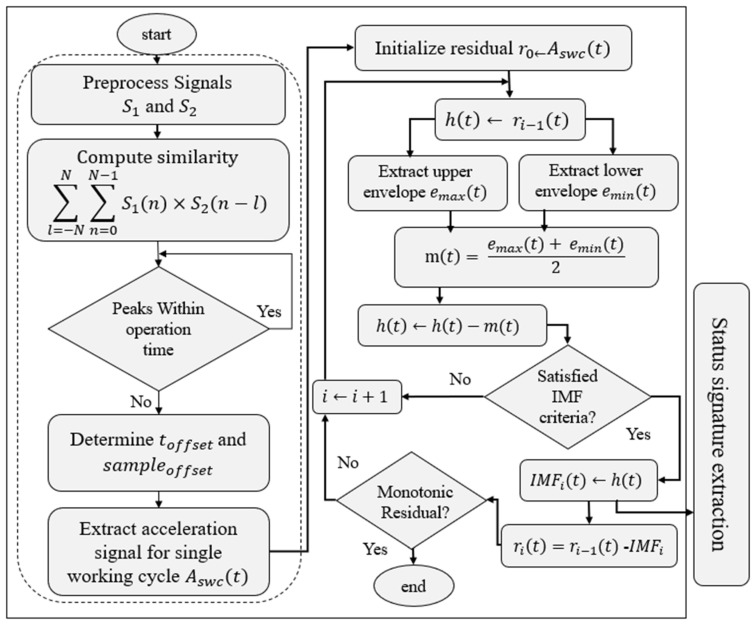
The block diagram of the proposed status signature extraction through cycle-informed segmentation.

**Figure 3 sensors-25-04431-f003:**
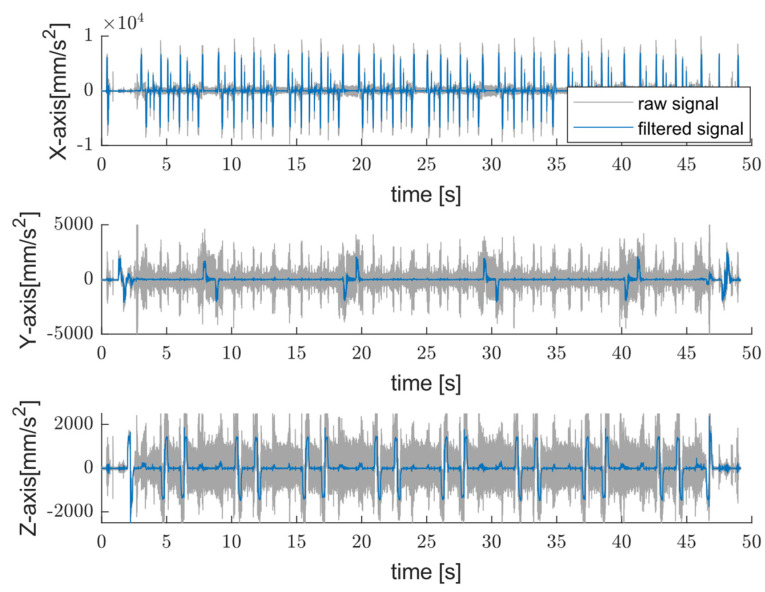
Example of time-domain acceleration [27] “https://github.com/boschresearch/CNC_Machining, accessed on 1 July 2022” and filtered signal (light blue trace) acquired along three axes.

**Figure 4 sensors-25-04431-f004:**
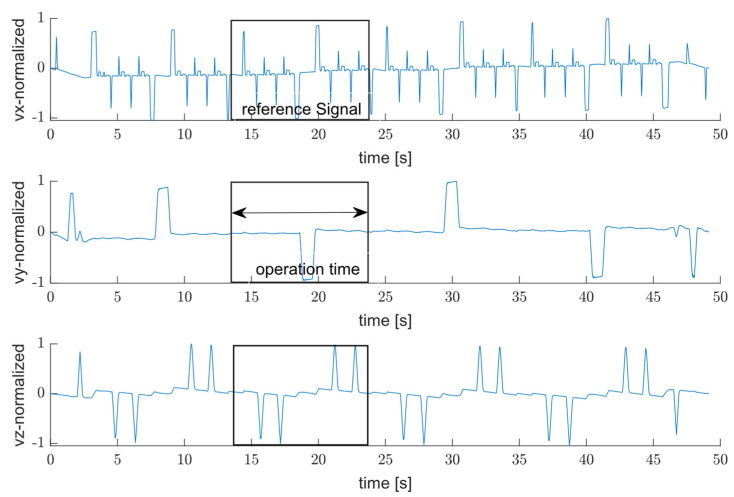
An example of extracted velocity profile along three orthogonal axes x, y, and z.

**Figure 5 sensors-25-04431-f005:**
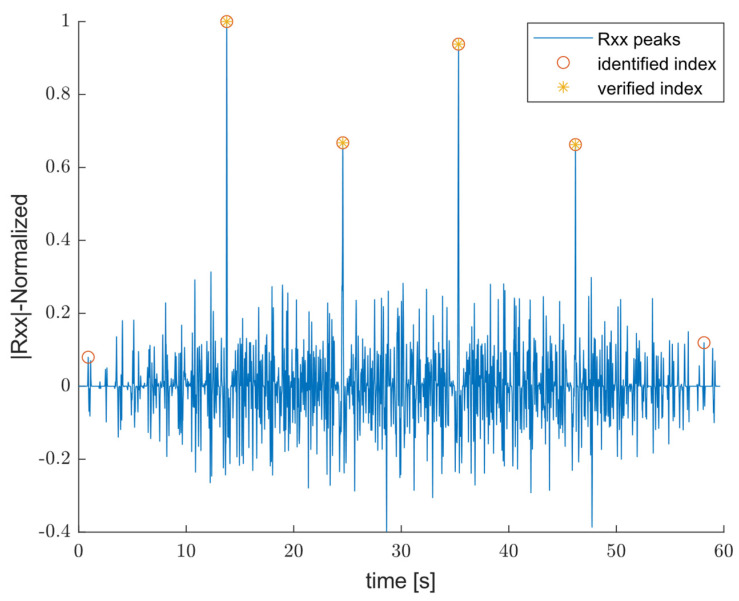
An example of normalized amplitude of the cross-correlation with identified and verified single-operating-cycle indices for signal segmentation.

**Figure 6 sensors-25-04431-f006:**
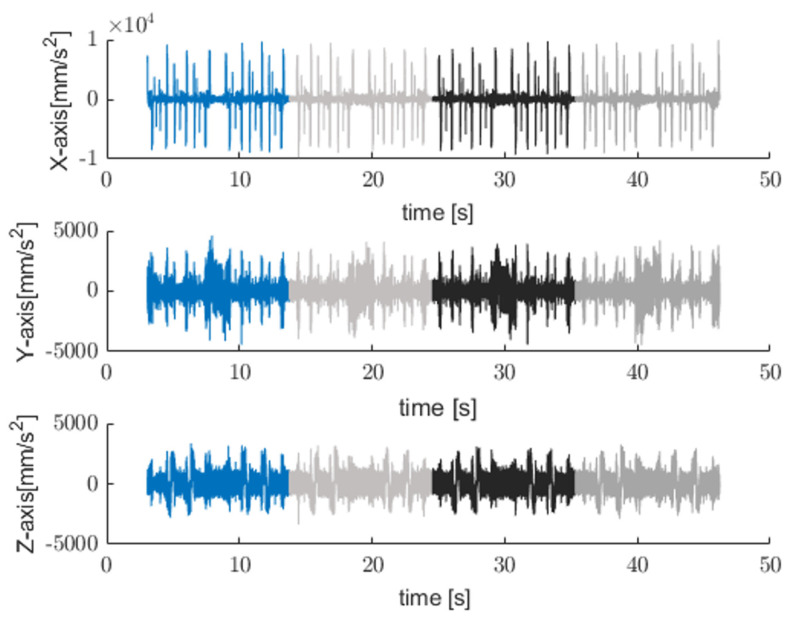
An example of segmented acceleration signals along three orthogonal axes associated with the single operation cycle using the proposed approach.

**Figure 7 sensors-25-04431-f007:**
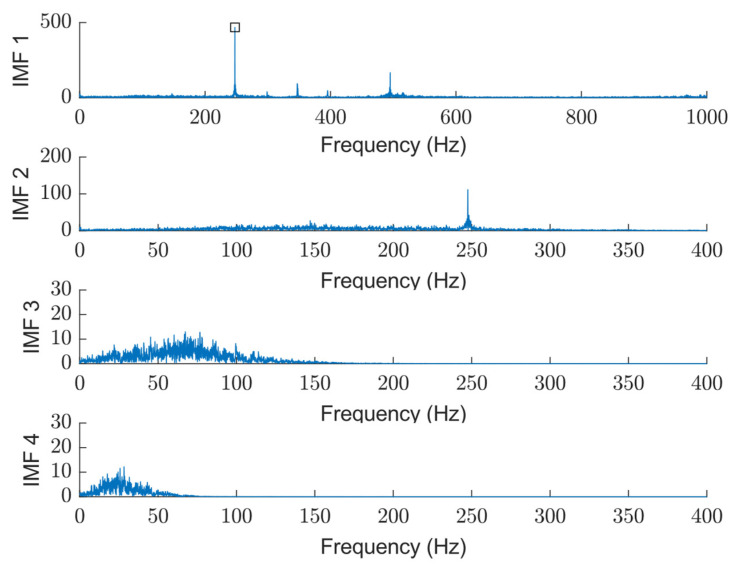
An example of a spectrum of extracted IMFs for single segmentation: resulted detected peak in IMF1 from the peak detection algorithm is highlighted in a square.

**Figure 8 sensors-25-04431-f008:**
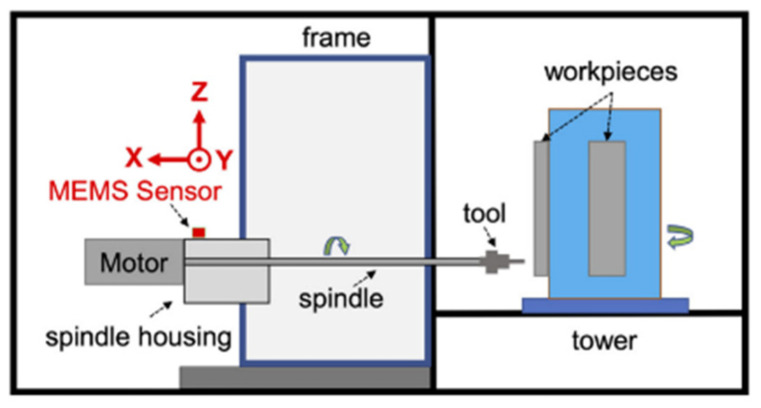
Schematic sketch of the experimental setup: 4-axis machining center with mounted sensor presented in [27].

**Figure 9 sensors-25-04431-f009:**
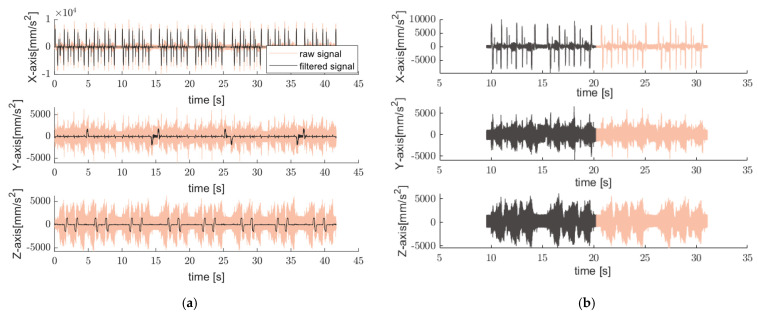
(**a**) Time-domain of full signal [28] and (**b**) segmented acceleration signals along three orthogonal axes associated with the single operation cycle using the proposed approach for OP10-NOK-process (light orange trace).

**Figure 10 sensors-25-04431-f010:**
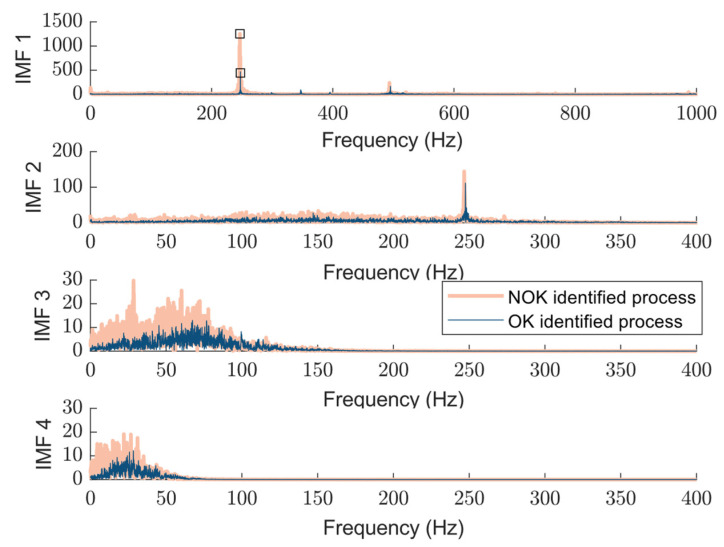
Comparison between extracted status signature using proposed approach while the machine executes a single working cycle of OP10 for annotated OK (dark blue) and NOK (light orange trace) processes: resulted detected peak in IMF1 from the peak detection algorithm is highlighted in a square.

**Figure 11 sensors-25-04431-f011:**
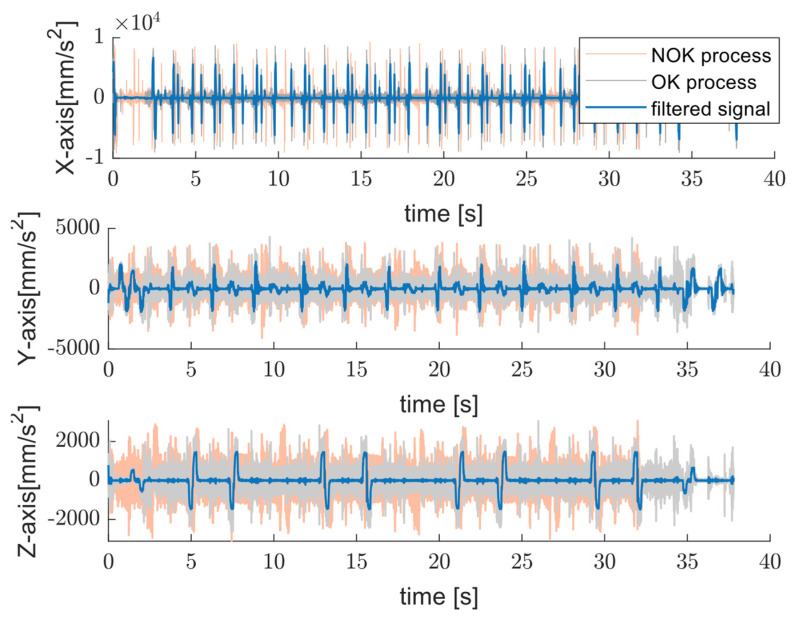
Time-domain acceleration [27] and filtered signals (blue trace) acquired along three orthogonal axes for OK (gray trace) and NOK processes (light orange trace) while operating OP08.

**Figure 12 sensors-25-04431-f012:**
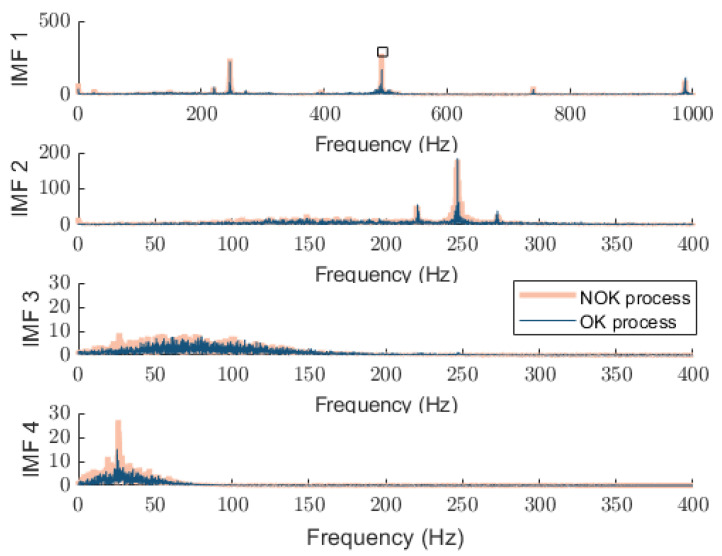
Distinguishable spectrum of IMFs using proposed approach while the machine executes a single working cycle of OP08 for OK (dark blue trace) and NOK processes (light orange trace): resulted detected peak in IMF1 from the peak detection algorithm is highlighted in a square.

**Figure 13 sensors-25-04431-f013:**
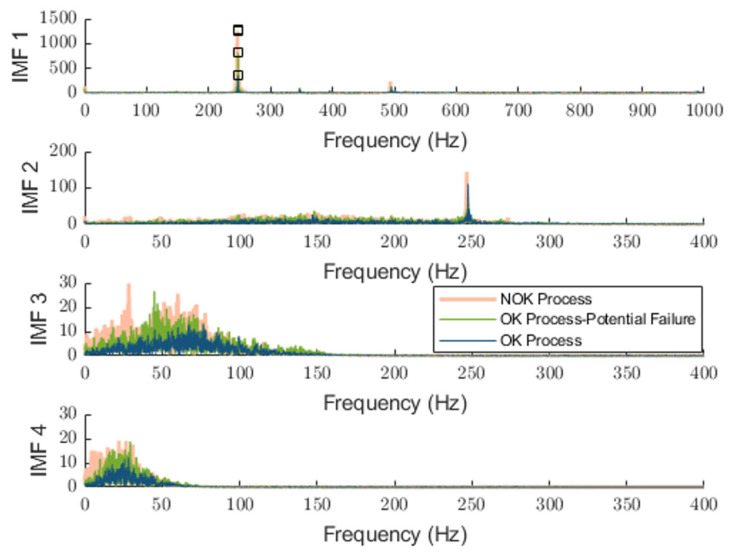
An example of extracted status signature addressing distinguishable, OK process (dark blue trace), potential to failure process (green trace), and NOK process (light orange trace) using proposed method: resulted detected peak in IMF1 from the peak detection algorithm is highlighted in a square.

**Figure 14 sensors-25-04431-f014:**
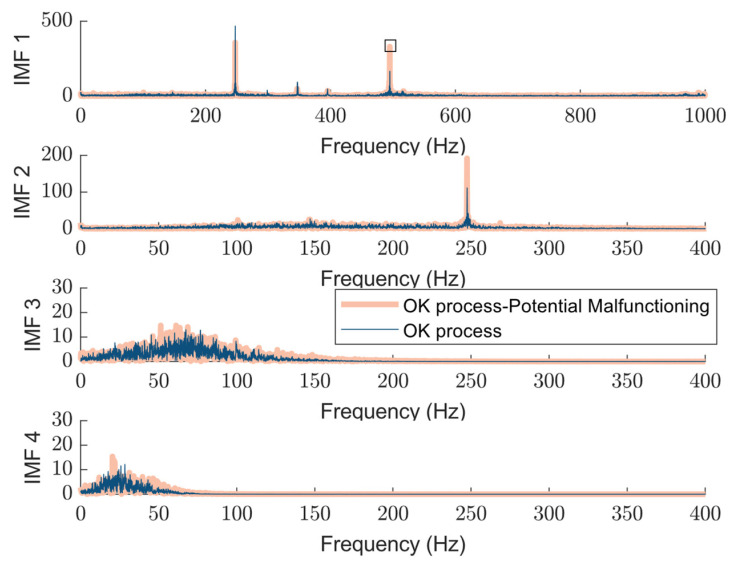
An example of extracted status signature addressing distinguishable potential tool malfunctioning using proposed method: OK process (dark blue trace), and Potential Malfunctioning (light orange trace): resulted detected peak in IMF1 from the peak detection algorithm is highlighted in a square.

**Table 1 sensors-25-04431-t001:** Case-study-based evaluation of proposed cycle-informed triaxial sensor.

Case Study	Context	Algorithm Result	Key Observation	Effectiveness
Fast Tool Change and Short Cycle Duration	One of the most challenging cases due to rapid operations and minimal segmentation margin	Dynamically and accurately identified physical cycles, avoiding signal overlapping and misalignment unlike fixed-window methods	Distinct harmonic content in resulting IMFs, allowing differentiation between OK and NOK cycles	Robustness in high-speed, low-margin processes with strong spectral clarity without heavy computational effort or additional sensors
Potential-to-Failure Transition	Expert-annotated borderline cases, hard to classify with binary labels only on final product quality	Gradual noise floor rise and broader harmonic components in higher-order resulting IMF Sideband growth on lower-order resulting IMFs	Provided intermediate diagnostic resolution to flag early risk without full-failure onset	Enabled detection of process drift and actionable precursors for predictive maintenance and waste-less production
Tool Wear Progression	Gradual degradation across cycles with intermediate “Potential Malfunction” class	IMF1 and IMF2 spectra showed emergence of second harmonics and sideband components	Amplified frequency spread and imbalance in harmonic ratios indicated early-stage tool wear	Captured subtle degradation patterns missed by static thresholding or windowed transforms

## Data Availability

The original contributions presented in this study are included in the article. Further inquiries can be directed to the corresponding author.

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
