# Peer review of "Cycle-Informed Triaxial Sensor for Smart and Sustainable Manufacturing"

_sensors, 2025, doi:10.3390/s25144431_

Round 1
Reviewer 1 Report
Comments and Suggestions for Authors
The main idea of the article is to use data from a single three-axis accelerometer and apply EMD to detect degradation and failures in CNC machines. The authors propose a cycle-synchronized signal segmentation, which eliminates the need to access the machine controller. The method offers a practically useful solution for monitoring CNC machines, but I have some comments on the article:
1) Do I understand correctly that this article is aimed at processing the dataset taken from [25] and you are not related to this data in any way? You did not make and test your own setup for data collection? It would be nice to test not only on one dataset. Testing only on one type of equipment (CNC milling machines) and 15 operations. It is unclear how the method scales to other machines (e.g. turbines or robots). The sampling frequency of 2 kHz may be insufficient for high-frequency defects (e.g. cracks in bearings).
2) I would like to see some numerical comparison of the proposed method, what is the accuracy of anomaly detection? How and by how much is your method better than others? There are articles that also used this dataset to detect anomalies (but using ML methods)
Small comments
1) Line 332. ...a two-stage learning method proposed in [28]. Incorrect link, should be [29]. And the link to line 337 is also incorrect accordingly. Check all links in the text.
2) It would be nice to add some articles on your topic:Classification of Bearing Faults by Approximation of Peak-To-Peak Amplitudes Distribution
High-resolution time-series classification in smart manufacturing systems
Deep Learning for Anomaly Detection in CNC Machine Vibration Data: A RoughLSTM-Based Approach
Author Response
Response to Reviewers for the paper Sensors-3674581
“Cycle-Informed Triaxial Sensor for Smart and Sustainable Manufacturing”
We wish to thank the Editor in Chief, the Associate Editor and the Reviewers for the comments, which helped to improve the quality of the paper.
Our responses to the comments are written in red.
Reviewer 1:
- Do I understand correctly that this article is aimed at processing the dataset taken from [25] and you are not related to this data in any way? You did not make and test your own setup for data collection? It would be nice to test not only on one dataset. Testing only on one type of equipment (CNC milling machines) and 15 operations.
- a) We appreciate your thoughtful comment and welcome the opportunity to clarify our methodological choices and vision.
We confirm that we were not involved in the creation or collection of the dataset referenced in [28]. Our motivation to selecting this dataset and avoiding laboratory based available datasets are listed in 3 key reasons:
- Long-term nature: It was acquired over two years from a real industrial CNC machine, making it one of the few public datasets that reflects real process variability and operational dynamics.
- Industrial realism and challenge: The dataset is challenging due to its noise, binary labeling, and lack of controller data—characteristics commonly found in brownfield or controller-limited industrial systems. This aligns well with our paper’s goal: to propose an interpretable, embedded-compatible method that works even under such constraints.
- Broader methodological purpose: The aim of this paper is not to optimize anomaly detection accuracy on a single dataset. Instead, we address a critical problem in industrial signal processing, namely, the misalignment and distortion caused by fixed-length segmentation of dynamic vibration signals. Our cycle-informed approach ensures that diagnostic features are extracted from physically meaningful units of operation and therefore integrated domain knowledge.
Actually, the key characteristic that we found in this dataset motivates us to work in parallel to create a very close dataset of our own, considering further generalization, which, of course, requires time.
To address this, we add the following paragraph in the manuscript: Section1 lines 328-335
The proposed methodology offers a unique area for addressing real world machining process research. By investigation on the publicly available industrial benchmark dataset [28], the performance of the methodology is validated on a practical basis. The key reason for selecting such benchmark lies in its long-term coverage and realistic variability features. In addition, binary labeling is provided based on the inspection of the expert which provides a rare ability to evaluate segmentation and diagnostic algorithms under limitations. Finally, while majority of available study are simulation-based or laboratory conditions studies, this benchmark reflects real operational values such as operational noise, tool wear and cycle diversity.
- b) It is unclear how the method scales to other machines (e.g. turbines or robots).
We appreciate your comment on this border prospective. To address it clearly, we added following paragraph in Section 1, lines 140-148
Although the proposed method is evaluated solely on CNC milling processes with a relatively low sampling rate of 2 kHz, it is broadly applicable to other industrial systems exhibiting periodic behavior. This comprises automated assembly lines, injection molding systems, and robotic arms used in pick-and-place tasks. In such cases, even without access to controller data, these systems generate repeatable mechanical cycles that produce characteristic vibration patterns. The proposed approach is well suited to monitoring these processes, particularly in brownfield or controller-limited environments where internal machine signals are unavailable, but vibration data can still be captured externally.
c)The sampling frequency of 2 kHz may be insufficient for high-frequency defects (e.g. cracks in bearings).
We acknowledge the sampling frequency limitation and clarify in the revised manuscript by adding the following paragraph: Section 4, lines 535-544
While the proposed method demonstrates strong potential for interpretable signal segmentation and fault signature extraction in industrial machining, certain limitations outline opportunities for future enhancement. The framework assumes a degree of cycle periodicity, which aligns well with many industrial operations but may require adaptation for processes with highly irregular timing or non-repetitive tool paths. The current sampling rate (2 kHz) is sufficient for process-level dynamics but may not capture micro-scale or high-frequency faults such as localized early stage bearing defects. Such further improvements can lead to a smart and sustainable digital twin enabler where these cycle-resolved features are integrated as a foundation for future machine learning models where fault tolerance and early failure detection are crucial.
- I would like to see some numerical comparison of the proposed method, what is the accuracy of anomaly detection? How and by how much is your method better than others? There are articles that also used this dataset to detect anomalies (but using ML methods)
Thank you for raising this important point. We fully agree that performance benchmarking is critical when evaluating machine learning-based classifiers. This paper does not introduce a classifier or propose a full anomaly detection pipeline. Instead, our contribution targets a preceding and foundational stage of the diagnostic workflow: Other studies using this dataset such as cited references [28] and [29] focus on training supervised ML models to classify between OK/NOK cycles. While valuable, these models often rely on fixed-length windows or assume precise manual segmentation, conditions that are rarely feasible in real production lines. Our method removes this bottleneck by introducing an embedded-compatible strategy that automatically aligns signals to physical machine cycles using only accelerometer data, without access to control logic or encoder feedback. This preserves causal and spectral integrity and eliminates ambiguities introduced by artificial segmentation, thus improving the interpretability and reliability of any downstream classification or prognosis. That is why we see our work not as a replacement for anomaly detection models, but as a critical enabler, making future anomaly detection systems more robust, explainable, and applicable in controller-limited industrial environments.
To address this, we add the following paragraph in the manuscript:
Section 2, lines 194-198
Instead of introducing another classifier, this paper, we concentrate on developing interpretable signal decomposition aligned with real process cycles. It laying the groundwork for creating more comprehensive, multi-class annotated datasets. Table 1 presents the pseudo-code of the proposed cycle-informed segmentation based status signature.
Section 3.2, lines 402-411
As noted earlier, most studies using this dataset train various machine learning models to distinguish between OK and NOK cycles. Although useful, these approaches typically depend on fixed-length windows or require exact manual segmentation. But these conditions are rarely practical in real-world production environments. Our approach overcomes this limitation by providing an embedded-friendly solution that automatically aligns signals with actual machine cycles. The proposed method tries to use only accelerometer data, without needing control logic or encoder inputs. This physically grounded, repeatable segmentation improves the accuracy and clarity of the extracted status signatures, creating a reliable foundation for future classification and machine learning applications.
Small comments
- Line 332. ...a two-stage learning method proposed in [28]. Incorrect link, should be [29]. And the link to line 337 is also incorrect accordingly. Check all links in the text.
Thank you for your comment. The modification has been applied, and reference numbering is cross-checked.
- It would be nice to add some articles on your topic:
1.Classification of Bearing Faults by Approximation of Peak-To-Peak Amplitudes 2.Distribution High-resolution time-series classification in smart manufacturing systems 3.Deep Learning for Anomaly Detection in CNC Machine Vibration Data: A RoughLSTM-Based Approach
We appreciate your suggestion to include these interesting references especially. To do this we added the following paragraph in Section1 lines 75-83 and we updated the reference numbering:
Recently, Artificial Intelligence (AI) and Machine Learning (ML) have become vital tools for enhancing malfunction signature analysis [21]. For example, advanced deep learning techniques like the RoughLSTM-based approach for detecting vibration anomalies [22] show how effectively neural networks can model temporal patterns in CNC signals. In [23], a large-scale AI benchmarking study evaluated 36 machine learning and deep learning models. These studies include convolutional networks like ResNet and InceptionTime, as well as recurrent networks like LSTM and BiLSTM. The findings demonstrate that deep convolutional models consistently deliver high classification accuracy. underscoring the expanding role of data-driven AI in smart manufacturing.
Following references are added in reference section
[21] Karimov, T.I.; Logunov, O.Y.; Druzhina, O.S.; Kolev, G.Y.; Kopets, E.E.; Kaplun, D.I. Classification of bearing faults by approximation of peak-to-peak amplitudes distribution. Proc. NIELIT Int. Conf. Commun. Electron. Digit. Technol. 2024, 3–13.
[22] Turan, A.;Çekik, R. Deep learning for anomaly detection in CNC machine vibration data: A RoughLSTM‑based approach. Appl. Sci. 2025, 15, 3179.
[23] Farahani, M.A.; McCormick, M.R.; Harik, R.; Wuest, T. Time‑Series Classification in Smart Manufacturing Systems: An Experimental Evaluation of State‑of‑the‑Art Machine Learning Algorithms. Robot. Comput. Integr. Manuf. 2024, 91, 102839.
Reviewer 2 Report
Comments and Suggestions for Authors
This manuscript introduces a lightweight, embedded-compatible vibration sensing framework that combines cycle-synchronized signal segmentation and Empirical Mode Decomposition (EMD) using only a single triaxial MEMS accelerometer, targeting predictive maintenance in controller-limited or brownfield CNC machines. The proposed framework is evaluated on a real-world benchmark dataset and demonstrates the ability to differentiate between healthy, faulty, and early-degraded processes. The work is timely, well-written, and highly relevant to Industry 5.0 and edge-enabled maintenance. However, improvements are required in reproducibility, performance benchmarking, and comparative discussion with existing methods.
- The concept of cycle-informed segmentation without requiring access to CNC control signals is both innovative and practically valuable. Combining this with EMD allows effective condition monitoring using minimal sensor data, aligning well with low-power, embedded system goals. However, the novelty is somewhat diluted due to the absence of comparative performance evaluation with other segmentation techniques (e.g., fixed-windowing, STFT, wavelet-based methods on the same dataset). Add a comparative performance section to demonstrate quantitatively how the proposed segmentation+EMD outperforms fixed-window or other signal decomposition baselines.
- While the algorithm is well described, no code, pseudo-code, or public dataset link is provided. Include algorithmic pseudo-code or a flowchart summarizing the segmentation and decomposition process. Cite the exact URL/DOI for the benchmark dataset used, as it is central to the manuscript.
- The paper uses a real-world dataset and examines frequency-domain characteristics of different intrinsic mode functions (IMFs) for fault detection. However, No machine learning classifier is used to assess classification accuracy (F1, precision, recall) using extracted features. No quantitative evaluation metric (e.g., signal-to-noise ratio, anomaly score) is reported. Add or reference a small-scale ML model (e.g., SVM or prototypical network) to classify OK/NOK/PF and report standard classification metrics.
- Quantify the improvements in segmentation fidelity or fault detection over prior baselines.
- Figures 3–13 provide helpful examples of signals, filtered outputs, IMF spectra, and fault patterns. However, Some figures are low-resolution or missing legends, especially IMF spectra comparisons (Figures 7, 9, 11, 12). Improve image resolution and clarity.
- Use color-coding or legends to distinguish OK/NOK processes in each plot.
- A dedicated discussion of limitations is missing.
-
Add a short section discussing Generalization to other machine types, Robustness under varying noise levels, and sensitivity to sensor placement or alignment errors.
- Language needs to be polished.
In addiiton,
1. The research gap and novelty part should be enhanced for better readability, and it should be added at the end of the introduction section.
2. In addition to Figure 1, the Experimental setup (Picture) needs to be added.
3. State the cut-off frequency used in the filter.
4. What kind of wavelet processing (Family, DB, order) used in signal processing needs to be explored?
5. From Figure 13, how have the potential malfunctions been identified? Peaks were identified at 350 Hz also. How the authors claim a malfunction at 500 Hz in IMF1.
6. A separate Discussion section is required to brief the research findings.
7. Enhance the conclusion with key findings.
The English could be improved to more clearly express the research.
Author Response
Response to Reviewers for the paper Sensors-3674581
“Cycle-Informed Triaxial Sensor for Smart and Sustainable Manufacturing”
We wish to thank the Editor in Chief, the Associate Editor and the Reviewers for the comments, which helped to improve the quality of the paper.
Our responses to the comments are written in red.
Reviewer 2:
- The concept of cycle-informed segmentation without requiring access to CNC control signals is both innovative and practically valuable. Combining this with EMD allows effective condition monitoring using minimal sensor data, aligning well with lowpower, embedded system goals. However, the novelty is somewhat diluted due to the absence of comparative performance evaluation with other segmentation techniques (e.g., fixed-windowing, STFT, wavelet-based methods on the same dataset). Add a comparative performance section to demonstrate quantitatively how the proposed segmentation+EMD outperforms fixed-window or other signal decomposition baselines.
We sincerely appreciate your acknowledgment of the main contributions of our work and your helpful suggestion about including comparative evaluations to better measure performance improvements. Our primary focus has been to address a deeper, often neglected issue in real-world predictive maintenance which is dynamically segment vibration signals in a way that accurately reflects their operational context, especially when data sources are limited. Traditional fixed-length segmentation methods typically ignore actual machine dynamics. This leads to misaligned or incomplete segments related to machine cycles. This misalignment can dilute diagnostic value and mix signals from different operating states. In much of the existing literature, segmentation is applied without ensuring alignment to real cycles. In this paper to address it directly, we referred to the practical case (OP08) where machine operation and tool changes specifically become issue in fixed-window scenarios. As we illustrate qualitatively in Section 3.2 (please see Figures 11 and 12), this can cause feature mixing, lower interpretability, and overlapping frequency components between the OK and NOK classes. Our approach tackles this problem by explicitly aligning signal segments with true physical cycles, leading to cleaner, causally consistent inputs derived using readily available domain knowledge. While this study does not present a quantitative benchmark comparison, the advantages of process-aware segmentation are clearly shown through clearer harmonic patterns, more stable IMF decompositions, and interpretable differences among normal, faulty, and transitional conditions.
To address this, we add the following paragraph in the manuscript:
Section 2, lines 194-198
Instead of introducing another classifier, this paper, we concentrate on developing interpretable signal decomposition aligned with real process cycles. It laying the groundwork for creating more comprehensive, multi-class annotated datasets. Table 1 presents the pseudo-code of the proposed cycle-informed segmentation based status signature.
Section 3.2, lines 402-411
As noted earlier, most studies using this dataset train various machine learning models to distinguish between OK and NOK cycles. Although useful, these approaches typically depend on fixed-length windows or require exact manual segmentation. But these conditions are rarely practical in real-world production environments. Our approach overcomes this limitation by providing an embedded-friendly solution that automatically aligns signals with actual machine cycles. The proposed method tries to use only accelerometer data, without needing control logic or encoder inputs. This physically grounded, repeatable segmentation improves the accuracy and clarity of the extracted status signatures, creating a reliable foundation for future classification and machine learning applications.
- While the algorithm is well described, no code, pseudo-code, or public dataset link is provided. Include algorithmic pseudo-code or a flowchart summarizing the segmentation and decomposition process. Cite the exact URL/DOI for the benchmark dataset used, as it is central to the manuscript.
Thank you for your valuable comment. We agree that reproducibility and transparency are critical, especially in industrial research. To address this: We include pseudo-code in this version of the paper as Table 1, Section 2 line 211-212.
Algorithm 1: Cycle-Informed Segmentation and EMD |
Function decompose((t)); |
In addition, the public dataset used in this study has now been explicitly cited using its link as mentioned by the authors:
[28] Tnani, M.A.; Feil, M.; Diepold, K. Smart Data Collection System for Brownfield CNC Milling Machines: A New Benchmark Dataset for Data-Driven Machine Monitoring. Procedia CIRP 2022, 107, 131–136. https://github.com/boschresearch/CNC_Machining.
We also updated the caption of Figure 3 where the dataset is first used, and figure 8 for clarity.
- The paper uses a real-world dataset and examines frequency domain characteristics of different intrinsic mode functions (IMFs) for fault detection. However, No machine learning classifier is used to assess classification accuracy (F1,precision, recall) using extracted features. No quantitative evaluation metric (e.g., signal-to-noise ratio, anomaly score) is reported. Add or reference a small-scale ML model (e.g., SVMor prototypical network) to classify OK/NOK/PF and report standard classification metrics.
Thank you for your thoughtful comment. We recognize the value of quantitative evaluation metrics, especially when assessing classification models. However, our goal here is not to benchmark a machine learning model like those presented in the additional references [21–23]. Our focus is to develop a robust and interpretable preprocessing approach for condition monitoring. As earlier studies using this dataset have shown [28, 29], classification performance is highly dependent on how segmentation and feature extraction are handled. Our framework addresses this by extracting complete, physically meaningful segments of vibration data. Then, it will apply EMD to generate rich, status-specific health signatures. These cycle-aligned features can then provide a solid basis for future machine learning models, particularly in cases with limited labeled data or machines exhibiting highly variable operating conditions.
To clarify this, we added this we updated the conclusion section:
- Quantify the improvements in segmentation fidelity or fault
We are competently agreeing that precise segmentation is crucial for vibration-based diagnostics, and we appreciate your suggestion. While not expressed in numerical fidelity metrics, resulting intrinsic mode functions as represented in this manuscript exhibit distinct and cycle-specific harmonic patterns that enable clearer differentiation between OK, NOK, and developing fault conditions. This qualitative improvement advances our broader goal of producing interpretable, physically meaningful diagnostic features while avoiding the misaligned or incomplete segments that are often overlooked in dynamic environments
- Figures 3–13 provide helpful examples of signals, filtered outputs, IMF spectra, and fault patterns. However, Some figures are low-resolution or missing legends, especially IMF spectra comparisons (Figures 7, 9, 11, 12). Improve image resolution and clarity.
We do appreciate your kind comment. You are absolutely right, the resolution of these figures are low. To solve this, we replaced these images with better quality in addition to updating the figure 8, 10,13.
- Use color-coding or legends to distinguish OK/NOK processes in each plot.
Thanks for your comment. Color code is used in all images for distinguishing OK and NOK process
- A dedicated discussion of limitations is missing.
We appreciate your suggestion and to address this we added the following paragraph in Section 5 line 535-544
While the proposed method demonstrates strong potential for interpretable signal segmentation and fault signature extraction in industrial machining, certain limitations outline opportunities for future enhancement. The framework assumes a degree of cycle periodicity, which aligns well with many industrial operations but may require adaptation for processes with highly irregular timing or non-repetitive tool paths. The current sampling rate (2 kHz) is sufficient for process-level dynamics but may not capture micro-scale or high-frequency faults such as localized early stage bearing defects. Such further improvements can lead to a smart and sustainable digital twin enabler where these cycle-resolved features are integrated as a foundation for future machine learning models where fault tolerance and early failure detection are crucial.
- Add a short section discussing Generalization to other machine types, Robustness under varying noise levels, and sensitivity to sensor placement or alignment errors.
Thank you for highlighting this important aspect, which helped us reflect more broadly on the applicability and practical robustness of the proposed method. To address this, we added the following paragraph.
Section 1, lines 140-148
Although the proposed method is evaluated solely on CNC milling processes with a relatively low sampling rate of 2 kHz, it is broadly applicable to other industrial systems exhibiting periodic behavior. This comprises automated assembly lines, injection molding systems, and robotic arms used in pick-and-place tasks. In such cases, even without access to controller data, these systems generate repeatable mechanical cycles that produce characteristic vibration patterns. The proposed approach is well suited to monitoring these processes, particularly in brownfield or controller-limited environments where internal machine signals are unavailable, but vibration data can still be captured externally.
Section 5, lines 531-544
In terms of robustness, the use of low-pass filtering and velocity-domain correlation makes the approach resilient to high-frequency noise and minor transient disturbances. Moreover, the method is tolerant to reasonable variation in sensor mounting orientation, as long as primary motion axes are captured.
In addition,
- The research gap and novelty part should be enhanced for better readability, and it should be added at the end of the introduction section.
Thank you for this valuable suggestion. To improve clarity and structure, we have revised Introduction to explicitly highlight the research gap and the core novelty of our contribution.
Section1, line 99-114
In this paper, our focus is to tackle a more fundamental and often overlooked challenge in real-world and dynamic predictive maintenance schemes. Traditional fixed-length segmentation approaches do not account for actual machine dynamics and often result in misaligned or cycle-incomplete segments. This diluting diagnostic relevance and potentially mixing signals from different operational states. In this paper, the core of the contribution lies in a cycle-synchronized segmentation method, which leverages velocity profiles extracted directly from accelerometer signal. Consequently, avoiding misinterpretation of dynamic vibration data and enabling a causally coherent view of each machining cycle. This addresses a critical research gap in the literature, where most vibration-based monitoring frameworks assume access to controller signals or static segmentation windows. Unlike existing methods, the proposed approach is designed to operate in controller-limited environments and enables the extraction of interpretable, physically grounded health signatures that can serve as reliable input for downstream classification or prognosis. This novelty is particularly relevant for embedded monitoring applications where cycle awareness, low computational overhead, and explainability are key requirements.
- In addition to Figure 1, the Experimental setup (Picture) needs to be added.
Thank you for this helpful suggestion. As the dataset used in this work is publicly available and was not acquired by us, we do not have access to original photographs of the experimental setup. However, to improve clarity and context for readers we added the schematic of the experimental setup as reported in public dataset to the manuscript and all figures are renumbered accordingly.
Figure 8. Schematic sketch of the experimental setup: 4-axis machining center with mounted sensor presented in [28].
It is worth mentioning our motivation to selecting this dataset.
Our motivation to selecting this dataset are listed in 3 key reasons:
- Long-term nature: It was acquired over two years from a real industrial CNC machine, making it one of the few public datasets that reflects real process variability and operational dynamics.
- Industrial realism and challenge: The dataset is challenging due to its noise, binary labeling, and lack of controller data—characteristics commonly found in brownfield or controller-limited industrial systems. This aligns well with our paper’s goal: to propose an interpretable, embedded-compatible method that works even under such constraints.
- Broader methodological purpose: The aim of this paper is not to optimize anomaly detection accuracy on a single dataset. Instead, we address a critical problem in industrial signal processing, namely, the misalignment and distortion caused by fixed-length segmentation of dynamic vibration signals. Our cycle-informed approach ensures that diagnostic features are extracted from physically meaningful units of operation and therefore integrated domain knowledge.
Actually, the key characteristic that we found in this dataset motivates us to work in parallel to create a very close dataset of our own, considering further generalization, which, of course, requires time.
- State the cut-off frequency used in the filter.
We thank the reviewer for this observation. The cut-off frequency used in the low-pass Butterworth filter was 50 Hz, which was selected to preserve process-level motion dynamics while attenuating high-frequency noise and structural resonance artifacts. This information has now been explicitly stated in Section 2.1, line 218 of the revised manuscript for clarity.
- What kind of wavelet processing (Family, DB, order) used in signal processing needs to be explored?
Thank you mentioning this consideration. The proposed method relies solely on empirical mode decomposition (EMD) to extract intrinsic mode functions (IMFs) from cycle-aligned vibration segments. Unlike wavelet transforms, EMD is fully data-driven and does not require pre-selection of a wavelet family, decomposition level, or mother function, therefore our focus was using adaptive, non-parametric techniques that are better suited for non-stationery signals. However, techniques such as Symlet families, which is widely adopted in vibration-based fault diagnosis can be explored in future.
- From Figure 13, how have the potential malfunctions been identified? Peaks were identified at 350 Hz also. How the authors claim a malfunction at 500 Hz in IMF1.
We do appreciate your insightful question. The identification of potential malfunctions is not based solely on the presence of isolated peaks but rather on the differential harmonic behavior across process states. In Figure 14, the process labeled as “OK with possible malfunction” exhibits a more pronounced second harmonic near 500 Hz in IMF1 compared to the baseline OK process. This shift in harmonic balance—specifically, the increased ratio of second to first harmonic—is indicative of nonlinear behavior and emerging dynamic instability, which are common precursors of tool malfunction (e.g., imbalance, misalignment, or onset of chatter).
- A separate Discussion section is required to brief the research findings.
We thank the reviewer for this valuable suggestion. In response, we have now added a dedicated Discussion section (Section 4) that summarizes the core findings of the study: Section 4, lines: 478-504
- Discussion
In this paper, we introduced a novel cycle-informed segmentation and IMF-based decomposition framework that enables meaningful and interpretable health status signatures from vibration data even in the absence of controller feedback, a condition commonly encountered in brownfield or controller-limited industrial systems. Many existing literatures apply segmentation without ensuring alignment to true machining cycles. This leads to misaligned or incomplete segments related to machine cycles causing non-realistic diagnostic value and mixed signals from different operating states. Com-pared to fixed-window methods, our approach consistently extracts cycle-resolved segments that better align with physical process boundaries, thereby preserving causal relationships and improving the reliability of extracted features. To effectively address such gaps, we focused on data from real-world scenarios where it reflects real process variability, operational dynamics, presence of noise, binary labeling on final quality, and lack of controller data, lack of historical data, characteristics commonly found in brownfield or controller-limited industrial systems. This aligns well with our paper’s goal: to propose an interpretable, embedded-compatible method that works even under such constraints. In Table.2, we summarized three most challenging case studies, and while evaluating the key observations and effectiveness of the proposed algorithm. As shown in this table, the proposed method demonstrated high sensitivity to evolving tool–workpiece interaction, with the ability to detect latent anomalies before they become critical. Furthermore, the framework remains lightweight and well-suited for embedded deployment. It requires no model retraining or parameter tuning and relies only on cycle-adapted signal filtering, integration, and cross-correlation making it compatible with resource-constrained platforms. This design supports controller-independent, scalable predictive maintenance systems and aligns with the emerging goals of Industry 5.0.
Table 2. Case Study-Based Evaluation of proposed Cycle-Informed triaxial sensor
Case study |
Context |
Algorithm Result |
Key Observation |
Effectiveness |
Fast Tool Change & Short Cycle Duration |
One of the most challenging cases due to rapid operations and minimal segmentation margin |
Dynamically and accurately identified physical cycles, avoiding signal overlapping and misalignment unlike fixed-window methods which |
Distinct harmonic content in resulted IMFs, allowing differentiation between OK and NOK cycles |
Robustness in high-speed, low-margin processes with strong spectral clarity without heavy computational effort or additional sensors |
Potential-to-Failure Transition |
Expert-annotated borderline cases, hard to classify with binary labels only on final product quality
|
Gradual noise floor rise and broader harmonic components in Higher order resulted IMF Sideband growth on lower order resulted IMFs |
Provided intermediate diagnostic resolution to flag early risk without full-failure onset. |
Enabled detection of process drift and actionable precursors for predictive maintenance and waste-less production |
Tool Wear Progression |
Gradual degradation across cycles with intermediate “Potential Malfunction” class. |
IMF1 and IMF2 spectra showed emergence of second harmonics and sideband components |
Amplified frequency spread and imbalance in harmonic ratios indicated early-stage tool wear. |
Captured subtle degradation patterns missed by static thresholding or windowed transforms. |
- Enhance the conclusion with key findings.
Thank you for the helpful suggestion. We have revised the Conclusion to include a clearer synthesis of the study’s main contributions and experimental outcomes. Specifically, the updated version highlights the successful extraction of cycle-resolved vibration signatures, the interpretability and compactness of EMD-derived features, and the method’s compatibility with embedded processing environments. The conclusion also reiterates the method’s potential for future integration into FPGA/MCU platforms and its suitability for controller-free fault detection in industrial scenarios.
Round 2
Reviewer 1 Report
Comments and Suggestions for Authors
The authors have taken into account all the comments and added the necessary clarifications. The article has become much better and clearer. Inaccuracies have been corrected and now the contribution of the article to the subject area is highlighted much more clearly
I believe that the article is ready for publication by MDPI
Reviewer 2 Report
Comments and Suggestions for Authors
Congrats to the authors.